# The Effect of Postprocessing on the Fatigue Properties of Ti-5Al-5Mo-5V-1Cr-1Fe Produced Using Electron Beam Melting

**DOI:** 10.3390/ma16031201

**Published:** 2023-01-31

**Authors:** Michał Karoluk, Karol Kobiela, Marcin Madeja, Robert Dziedzic, Grzegorz Ziółkowski, Tomasz Kurzynowski

**Affiliations:** Faculty of Mechanical Engineering, Centre for Advanced Manufacturing Technologies (CAMT-FPC), Wroclaw University of Science and Technology, Lukasiewicza 5, 50-371 Wroclaw, Poland

**Keywords:** PBF-EB/M, Electron Beam Melting, EBM, Ti55511, fatigue performance, HIP

## Abstract

Despite the significant potential advantages of processing Ti-5Al-5Mo-5V-1Cr-1Fe alloy (Ti-55511) using Electron Beam Melting (PBF-EB/M), when compared to conventional manufacturing technologies, the resulting internal defects are an important characteristic of such additive technologies and can highly decrease mechanical properties. One of the most dangerous defects formed during metal additive manufacturing processes are material discontinuities such as a lack of fusion. Defects of this type, due to their “flat” nature, are difficult to characterize. For cycle-loaded specimens, where the loading force acts perpendicular to the lack-of-fusion plane, defects of this type can significantly reduce fatigue properties. This paper presents the results of research aimed at improving the fatigue properties of Ti55511 alloy by reducing the influence of the lack-of-fusion defect on fatigue damage. The static and fatigue properties of specimens in the as-built state, as well as after hot isostatic pressing (HIP) treatment, were analyzed. The effect of HIP on both the reduction of pores and the degree of sphericity when using the X-ray computed tomography (XCT) system was presented. The change in the microstructure after HIP was analyzed in terms of the change in the size of individual phases, as well as the change in the phase ratio. This paper also contains a fractographic analysis of the samples after tensile and fatigue tests.

## 1. Introduction

In recent years, additive manufacturing (AM) has found more and more applications in various fields such as the medical industry and—most notably—the aerospace industry [1,2,3]. The growing interest in these technologies is due to the fact that they make it possible to produce parts with complex geometries that are difficult or even impossible to produce using traditional cavity manufacturing processes. One such technology is Electron Beam Melting (PBF-EB/M) [1,2,3], which uses a highly focused electron beam with a high energy density to selectively melt metallic powder in a layered fashion. This ensures that only the material that is actually needed to make the components is used to produce net shape parts. As a result, parts can be produced at a highly efficient buy-to-fly ratio [4,5,6].

In addition, as the PBF-EB/M process is a hot process, i.e., it is carried out at higher temperatures, there is a minimization of the formation of the residual stresses that occur in processes with rapid cooling. Moreover, among the available AM technologies, the PBF-EB/M process is carried out under vacuum [5], which can be seen as a positive aspect when processing materials with a high affinity for oxygen. This is because such process conditions are advantageous when processing materials that are highly oxidizable and prone to fracture—such as titanium alloys [4,7,8].

Ti-5Al-5Mo-5V-1Cr-1Fe (Ti-55511) alloy is widely used in the aerospace industry. It is used for key structural components due to its high strength-to-density ratio, good hardenability, and excellent fatigue crack propagation properties [9]. Ti-55511 is a two-phase (α + β) transition (near-β) alloy that was one of the first of its type to be widely used [10,11]. It is an incandescent–strength alloy that is designed for plastic processing, and is used for parts that are subjected to high mechanical loads, i.e., chassis components, fuel system parts, or aircraft engine components (low-pressure compressor blades) [12].

An important aspect with regard to using Ti-55511 alloy in PBF-EB/M technology seems to be the fact that the processing temperatures are very close to those of one of the commonly used heat treatments for β-phase and transition (β-near) alloys—BASCA—(β Annealing, Slow Cooling, and Aging) [13].

Despite the significant potential advantages of processing Ti-55511 using PBF-EB/M technology when compared to conventional manufacturing processes, an important factor that is typical of additive technologies, and which affects the mechanical properties—that is, internal porosity—cannot be overlooked. Processing with the use of hot isostatic pressing (HIP) allows the removal/reduction of internal porosity, which consequently changes the mechanical properties to more favourable ones [14,15].

For this reason, the further use of Ti-55511 after the PBF-EB/M process requires the determination of the basic mechanical properties [13,16] and fatigue properties. It is worth noting that fatigue properties can be seen as one of the most significant parameters affecting the determination of the potential suitability of using this alloy in new aerospace applications.

The purpose of this study was to identify the factors that affect the high-cycle fatigue properties of the Ti-55511 alloy parts made using PBF-EB/M technology (in the as-built state, as well as after the HIP process), and to also determine how they do so.

## 2. Materials and Methods

The research to identify the factors and determine the relationship that affects the high-cycle fatigue properties of the Ti-55511 alloy parts made using PBF-EB/M technology was carried out according to the scheme shown in Figure 1.

The powder used for this study was formed after the plasma atomization process (APS) (Figure 2), and is characterized by having good sphericity and sparse adhered satellites—characteristic of APS technology. The performed cross-sectional scan of the powder showed no internal pores, no non-metallic inclusions, and no other material defects (Figure 2b).

The size of the powder was determined using laser diffraction (Figure 3), as well as dynamic image analysis using the Q3 (volume-base diameter) cumulative distribution according to ISO 13320:2009. The size distribution of the obtained powder for D10.3, D50.3, and D90.3 was 49 µm, 72 µm, and 109 µm, respectively.

For the standard layer thickness of 50 µm used in EBM, the preferred powder size is between 45 and 106 µm. After the process of melting the cross-section of the scanned layer, the free space between the powder particles is reduced, which in turn causes a collapse. The shrinkage phenomenon resulting from this fact causes the effective thickness of the applied layer to be 80 µm [18]. This means that a powder with a particle size in the range of 45–106 µm is suitable for nano-application of a layer with a theoretical thickness equal to 50 µm. The reference chemical composition of the powder according to the GOST standard [10], and that which is declared by the powder’s manufacturer, are presented in Table 1.

Specimens (Figure 4) for the tensile test (conforming to ASTM E8/E8M-22) and fatigue test (conforming to ASTM E466) were fabricated on an EBM Arcam A1 (ARCAM AB, Mölndal, Sweden). They were fabricated directly on a stainless steel platform in a vertical orientation (rotational axis in line with the construction direction). The fabrication process was carried out under reduced pressure (2.1 × 10^–3^ mbar). The surface temperature of the powder was maintained at 750–800 °C. The process parameters used to fabricate the samples were developed based on previous studies and were as follows: beam current ~10 mA, melt scan speed ~6000 mm s^−1^, speed function—86, accelerating voltage—60 kV, powder layer thickness ~50 µm, and zig-zag scanning strategy.

Both the specimens for the static tensile test (10 pieces) and the specimens for the fatigue test (50 pieces) were divided into two groups. The first group consisted of specimens in the as-built state, while the second group contained specimens treated using post-process hot isostatic pressing (HIP) in order to reduce their internal porosity [14]. The HIP process was carried out in a protective atmosphere of argon gas. The process parameters were selected on the basis of literature analysis for studies conducted using the HIP processing of Ti6Al4V [19,20] alloy, while at the same time taking into account differences in the onset temperatures of the recrystallization of the alloys.

The following values were adopted for the HIP processing: a temperature of 820 °C, a processing time of 2 h at the set temperature, a pressure of 150 MPa, and a cooling rate of 600 °C/1 h. A temperature that was 10 °C lower than the starting value of the onset of recrystallization for the Ti55511 alloy of $T_{\alpha +\beta \to \beta }^{ps}-830\;{}^{\circ}C$ was selected.

Both groups of specimens (as-built and HIP) were then subjected to turning, shaping, and grinding in order to improve the final surface quality and dimensions.

The microstructure observations were conducted using a ZEISS EVO 25 scanning electron microscope (CARL ZEISS, Oberkochen, Germany). Images were taken with a BSE detector with an acceleration voltage of 20 kV. Phase analysis was performed at the cross-section perpendicular to the build direction using a MiniFlex 600 X-ray powder diffractometer (XRD) with a HyPix stripe detector and Smart Lab Studio II software (Rigaku, Tokyo, Japan). The XRD measurement was conducted with Cu Kα radiation and a scanning speed of 4 °/min in the range of 30 °–90 °. The specimens for testing were prepared in accordance with the standard metallographic procedure. Finally, the polished surface was etched with Kroll’s reagent (2% HF, 5% HNO_3_, and 93% H_2_O).

A Zwick-Roell ZHVμ-A hardness tester (Zwick-Roell Group, Ulm, Germany) was used for the hardness characterization. The hardness test was carried out using the Vickers method in accordance with the ISO EN ISO 6507 standard for a load of 1 kg (HV1). The average measurement result was obtained by making 10 indentations on each group of specimens.

The defect analysis was carried out using the XCT system (METROTOM 1500, Carl Zeiss, Oberkohen, Germany) with the maximum accelerating voltage of the X-ray tube of 225 kV and a maximum resolution of more than 7 µm. The system is equipped with a flat-panel detector with a resolution of 1024 × 1024 pixels and a pixel size of 400 µm. The XCT system was used to analyze the porosity of the samples in the as-built state and also after the HIP process. The scan parameters for the XCT analysis are summarized in Table 2.

To describe the shape of the registered defects, the sphericity factor was used according to Equation (1), for which the shape of the sphere takes the value of the factor *S* = 1 [21].
(1)$$S=\frac{\pi^{1/3}{\left(6V\right)}^{2/3}}{A}$$
where *V* is the volume of the defect and *A* is the diameter of the sphere.

The tensile tests, including a static tensile test, were carried out on an INSTRON 3384 testing machine (INSTRON, Norwood, OH, USA). A 150 kN (2250 lb) load cell with an accuracy class of 0.5% and a non-contact video strain gauge (AVE 2663-821) was used for the tests. The tests were conducted at room temperature with a crosshead speed of 1 mm/min, and were continued until the sample broke. For each series, five measurements were carried out.

The fatigue tests were carried out on an Instron 8872 servohydraulic testing machine (INSTRON, Norwood, OH, USA). A 25 kN load cell was used for the tests. The specimens were tested with a tension–tension sinusoidal loading cycle, and with a constant stress ratio of (maximum stress/minimum stress) R = 0.1 and a frequency of 50 Hz. For each measurement series, the tests were carried out for five stress levels consisting of five measurements, with the effect of σ on the number of cycles to failure being determined. The result was plotted on a stress vs. cycles to failure curve (S-N). The fatigue tests were conducted for up to 10^7^ cycles.

## 3. Results and Discussion

### 3.1. Microstructure Characterization

Material processing in AM technologies involves, in many cases, changes in the chemical composition of the material. These changes result from the local delivery of a large amount of energy to a small volume of material, which results in the evaporation of alloying elements. Depending on the amount of energy supplied [22] and the conditions prevailing during the AM manufacturing process, the changes in the chemical composition may be negligible or may significantly affect the properties of the resulting material.

In the case of processing Ti55511 alloy using PBF-EB/M technology, the change in the chemical composition depends mainly on the scanning speed, the increase of which limits the evaporation of alloying elements: aluminium, molybdenum, and vanadium. The evaporation of aluminium in titanium alloys processed using PBF-EB/M is a common problem [17]. Table 3 shows the results of the XRF analysis of the processed Ti-55511 alloy.

The phase analysis carried out on the samples showed the presence of α and β phases (Figure 5). The quantitative Rietveld analysis performed on the obtained diffractograms showed that the content of the α phase in the as-built state is 44.3%. For the HIP state, the α content increased to 82.7%.

Figure 6 and Figure 7 show the microstructures of the Ti-55511 alloy specimens obtained by PBF-EB/M processing in the as-built state (Figure 6a) and after HIP treatment (Figure 6b). Analyzes of the light microscope images (Figure 6) show that the samples have a lamellar structure inside the primary β columns. Columnar grain growth occurred according to the direction of heat conduction. The diameter of the columns visible in Figure 6a,b are similar. For both samples, the α phase (αGB) is visible at the borders of the columns (red arrows), but significantly thicker in the HIP sample.

The SEM analysis (Figure 7) showed that inside the prior β columns there is a Widmanstatten-type structure consisting of α and β phase lamellae, which is typical for AM processed Ti-55511 alloy. The microstructure of the sample in the post-HIP state (Figure 7b) has noticeably darker α-phase lamellae than the sample in the as-built state (Figure 7a). This is confirmed by the phase composition results (Figure 6). The values calculated based on the XRD measurment (using the Rietveld analysis) showed that the HIP specimens have much more α phase than the as-built specimens (82.7% and 44.3%, respectively). A direct comparison of the two microstructures showed that the as-built sample exhibits greater anisotropy in terms of plate size. There are clearly darker areas (with a finer structure) and lighter areas (having thicker lamellae).

The hardness on the plane parallel (ZX) to the build direction was measured for the tested series in the as-built state and after HIP treatment. The measured hardness for the as-built samples was 390 ± 9 HV1. The hardness measured on the HIP samples was 359 ± 13 HV1.

### 3.2. Defect Characterization

Porosity was determined for each of the specimens in the same way for the selected region of interest (ROI), which included the gauge range of the specimen. The reconstruction results for the as-built and HIP specimens are shown in Figure 8.

The specimens in the as-built state had pore sizes larger than the XCT imaging resolution, with a density of >99.9%. For the HIP specimens, no pores with diameters larger than the applied measurement resolution were registered. The measured density values for the analyzed mechanical test samples are shown in Table 4.

To describe the spherical shape of the registered defects, the sphericity coefficient was used according to Equation (1). The dependence of the measured diameters of the observed defects as a function of the sphericity coefficient of the selected samples is shown in Figure 9.

An analysis of the shape of the defects using XCT showed that the diameter of most of the observed defects is smaller than 120 µm, while at the same time being characterized by a high sphericity coefficient of S > 0.5. The observed defects bear the signs of pores of a gaseous origin formed as a result of the evaporation of alloying elements during the PBF-EB/M process in a vacuum environment [23]. The HIP process made it possible to eliminate that porosity, which is a residue of the manufacturing process.

Furthermore, the analyzes of the light microscope images of cross-section along the build direction (Figure 10) confirmed the existence of gas pores observed in the XCT study in the as-built sample. Moreover, the lack-of-fusion defects between the layers were also identified. No visible defects were observed in the HIP sample.

### 3.3. Mechanical Properties

Figure 11 shows the stress-strain curves obtained in the tensile tests of two series of specimens: as-built and HIP. Based on the collected results of the measurements of mechanical properties after static uniaxial tensile testing, the basic statistical measures—mean and standard deviation—were calculated.

The values of the dispersion of the measurement series were analyzed in more depth by determining the coefficient of variation and the range. After HIP treatment, there was a decrease in tensile strength and yield strength; however, an increase in elongation was also observed (Table 5). The average value of ultimate tensile strength (UTS) was 1060 ± 19 MPa for the as-built specimens and 856 ± 5 MPa for the HIP specimens (Table 5). These values were slightly higher than the yield strength (YS), which was 1009 ± 24 MPa and 815 ± 8 MPa, respectively. This indicates a low hardening capacity, similar to that of conventionally as-built Ti55511 alloy. The largest coefficient of variation value was obtained when analyzing the strain values. The samples in the as-built state were characterized by an elongation (ε) of 12.2 ± 2.8%, with a large range value within the test series ranging from 8.7% to 15.4%. In contrast, the HIP specimens were characterized by an elongation equal to 17 ± 1.4%, with the range of the value being 55% lower when compared to the specimens in the as-built state (Figure 12).

The HIP specimens show a coefficient of variation about three times lower than the elongation values of the samples in the as-built state. W-testing (Shapiro–Wilk) was also performed for each measurement series to test the normality of the samples. For all the analyzed series, the obtained *p*-value of the test is greater than the assumed level of significance (α = 0.05), which does not allow us to reject the hypothesis that the sample comes from a normal distribution.

Figure 13 shows the surface of fractures after the tensile tests. The fracture surfaces showed similar dimples, indicating that all fractures occurred during uniaxial tensile loading. Figure 13a–c shows a specimen in the as-built state; internal defects are visible on the fractures, while no defects are visible on the specimen after HIP treatment (Figure 13d–f). Apart from the mentioned visible internal defect, the fracture surfaces of the specimens did not differ significantly.

HIP process in the case of two-phase titanium alloys is a procedure that significantly affects the final mechanical properties of the alloy. For most HIP process equipment, the ability to control the cooling rate is severely limited. Long cooling times at a temperature close to the T(α + β→β) transformation result in significant growth of the plastic alpha phase amount, which translates into significant changes in phase composition and a consequent decrease in UTS and YS while increasing elongation.

Figure 14 presents S-N curves showing the relationship between the applied stress and the number of cycles to failure for the as-built and HIP specimens. The S-N curves include regression lines for both the as-built and HIP specimens. Also included is the 95% confidence level for the regression fit (dotted lines). The fatigue limit, defined as the run-out stress at 10^7^ cycles, is 300 MPa for the as-built specimens. The HIP specimens, however, showed 50% improvement (fatigue strength at 10^7^ cycles is 450 MPa). The scatter of results is definitely greater for the specimens in the as-built state than for the specimens after HIP treatment.

Figure 15 shows microscopic images of the fatigue fractures representing the series for the maximum stress value, i.e., 500 MPa for the as-built specimens and 650 MPa for the HIP-treated specimens.

Flat defects, oriented perpendicular to the build direction, were observed at the fractures of all the as-built specimens. Such defects were mainly revealed at a short distance from the edge of the specimen. For each of the specimens, these were fatigue crack initiation sites. For the HIP-treated specimens, it was observed that crack initiation occurred inside the specimen and propagated radially outward. This shows that the HIP process changed the fracture characteristics of the Ti-55511 alloy parts produced using EBM technology and which were subjected to cyclic loading.

## 4. Conclusions

This article presents an evaluation of Ti-55511 alloy manufactured using PBF-EB technology, and investigates the effects of hot isostatic pressing on its microscopic and mechanical properties. To evaluate the correlation of microstructure, density, and internal defects with the fatigue behavior of the specimens, the material was evaluated in two states: in the as-built and HIP condition. The following main conclusions can be drawn:The microstructure of Ti-55511 manufactured using PBF-EB is of the Widmanstatten-type and consists of α and β phase lamellae inside primary β columns that grow according to the build direction.The HIP process causes significant changes in phase composition. It caused a significant growth of fine α phase separations from the as-built state, in turn increasing the amount of observed α phase by 40%—from 44.3% (as-built) to 82.7% (HIP).Analysis of the chemical composition before and after the HIP process showed no significant differences. The HIP process had no effect on the change in chemical composition. This eliminated one of the variables that could affect mechanical properties. Nevertheless, one cannot overlook the fact that after the EBM process, significant evaporation of aluminium was registered, which was also confirmed in paper [14].Manufactured specimens in the as-built state were characterized by a significant number of fine spherical pores. However, based on the analysis of fatigue fractures, it was observed that the main cause of fatigue cracks in the as-built specimens was flat defects known as a lack of fusion. These defects acted as the crack initiation site in the as-built specimens. Despite the high resolution of the CT scan, lack-of-fusion type defects were not detected. The HIP process effectively eliminated spherical pores and the effect of lack-of-fusion defects on the fatigue fracture.The HIP specimens were characterized by a higher elongation value and lower UTS, YS values when compared to the as-built specimens. Moreover, for all the measured values of mechanical properties, the range of the obtained results was significantly lower.For the HIP specimens, a 50% higher fatigue strength at 10^7^ cycles was observed (450 MPa) when compared to the as-built specimens (300 MPa).

## Figures and Tables

**Figure 1 materials-16-01201-f001:**
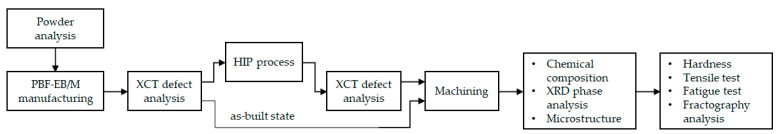
Overview of the methodology.

**Figure 2 materials-16-01201-f002:**
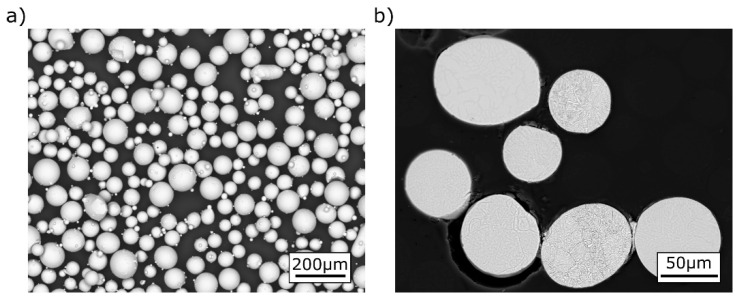
SEM/BSD images of Ti-55511 powder: (**a**) general view; (**b**) cross-sections [17].

**Figure 3 materials-16-01201-f003:**
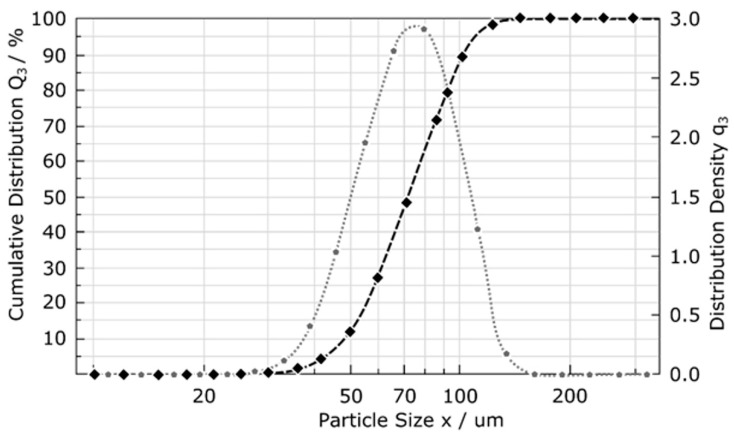
Analysis of the size distribution of the Ti-55511 powder fractions; dotted line—cumulative distribution, dashed line—distribution density [17].

**Figure 4 materials-16-01201-f004:**
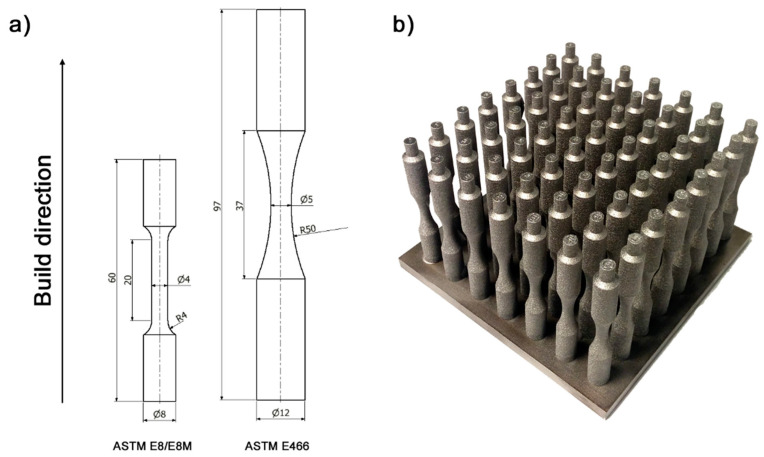
Specimens used in the studies: (**a**) geometry of the specimens for the tensile test (ASTM E8/E8M) and the fatigue test (ASTM E466), (**b**) manufactured PBF-EB/M—build layout.

**Figure 5 materials-16-01201-f005:**
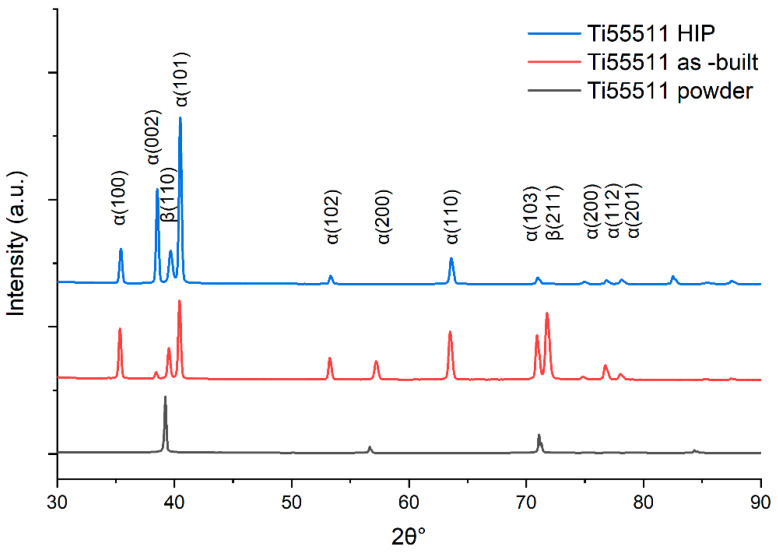
Diffractogram of the Ti-55511 specimens in the as-built and HIP state.

**Figure 6 materials-16-01201-f006:**
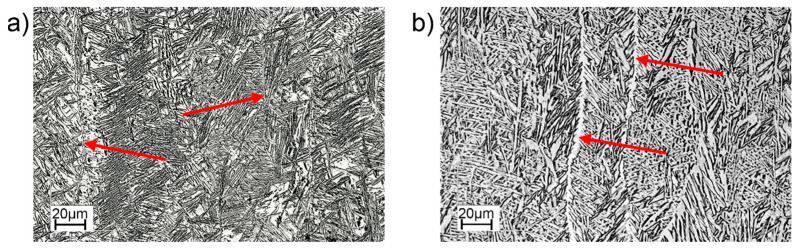
Microstructure of PBF-EB/Ti-55511: (**a**) as-built, (**b**) HIP. Borders of the column (αGB) marked by red arrows. Optical microscope.

**Figure 7 materials-16-01201-f007:**
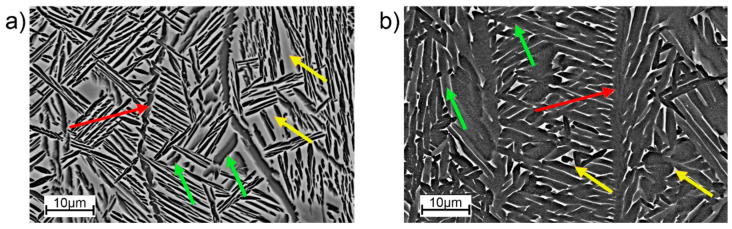
Microstructure of PBF-EB/Ti-55511: (**a**) as-built, (**b**) HIP. SEM. Arrows: red—borders of the column (αGB), yellow—β phase lamellae, green—α phase lamellae.

**Figure 8 materials-16-01201-f008:**
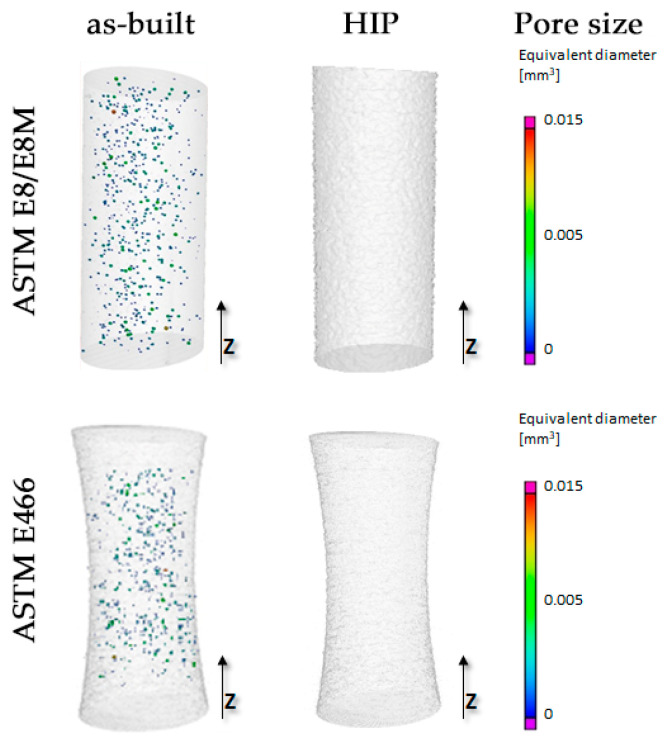
Specimens for the mechanical testing after CT reconstruction in the as-built and HIP state.

**Figure 9 materials-16-01201-f009:**
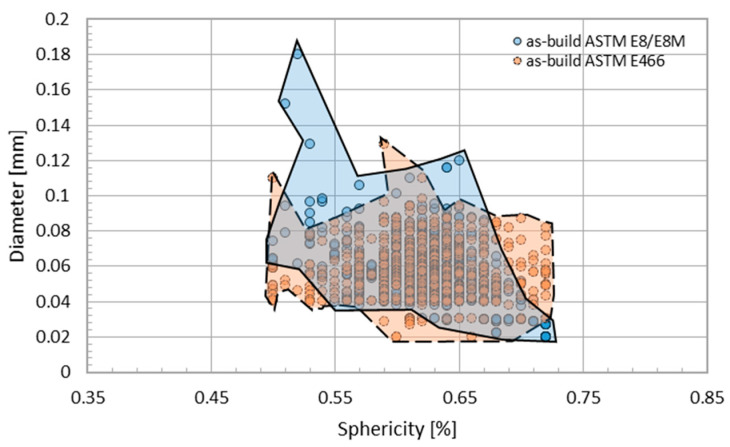
Sphericity and pore diameters in the as-built specimens.

**Figure 10 materials-16-01201-f010:**
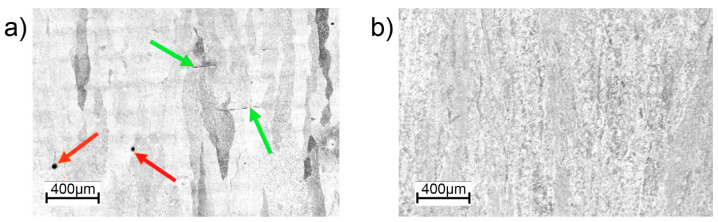
Microstructure of PBF-EB/Ti-55511: (**a**) as-built, (**b**) HIP. Optical microscope. Arrows: red—gas pores, green—lack-of-fusion defect.

**Figure 11 materials-16-01201-f011:**
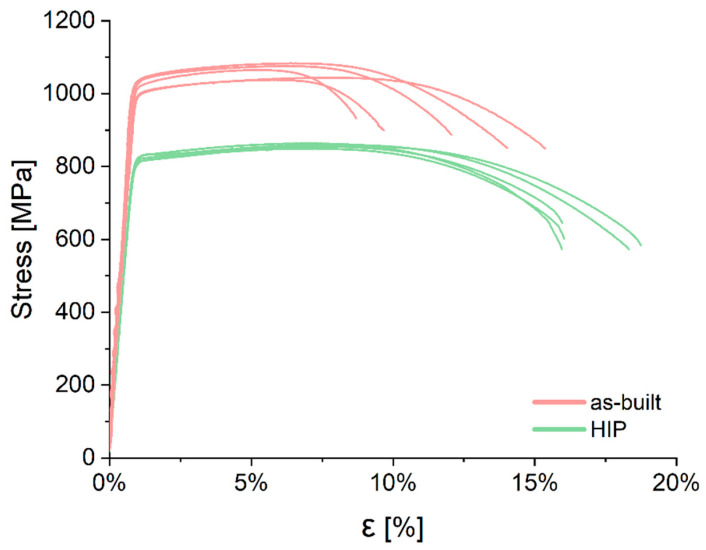
Tensile curves, comparison between the as-built and HIP specimens.

**Figure 12 materials-16-01201-f012:**
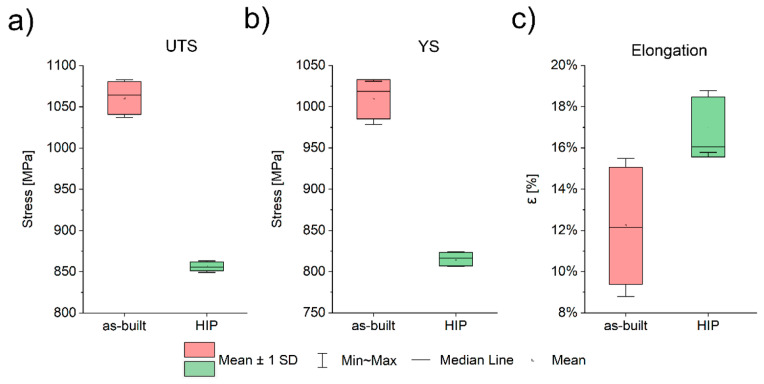
Tensile properties of the as-built and HIP state: (**a**) UTS, (**b**) YS, (**c**) elongation.

**Figure 13 materials-16-01201-f013:**
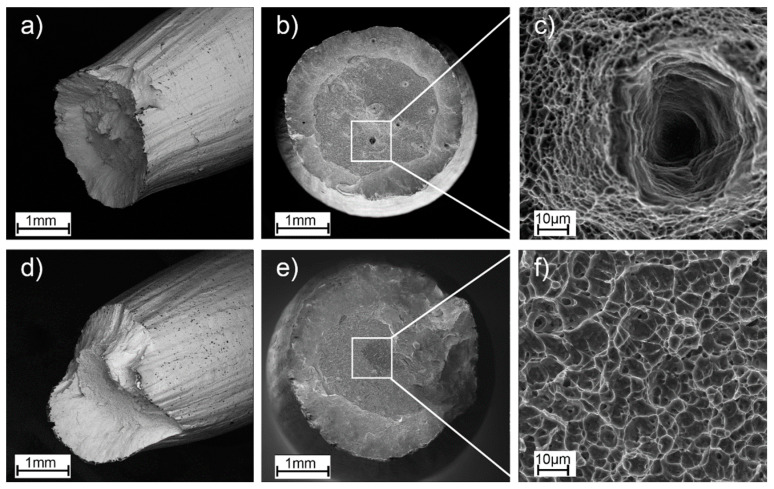
Fracture surfaces of the tensile-fractured specimen, (**a**–**c**) as-built, and (**d**–**f**) HIP.

**Figure 14 materials-16-01201-f014:**
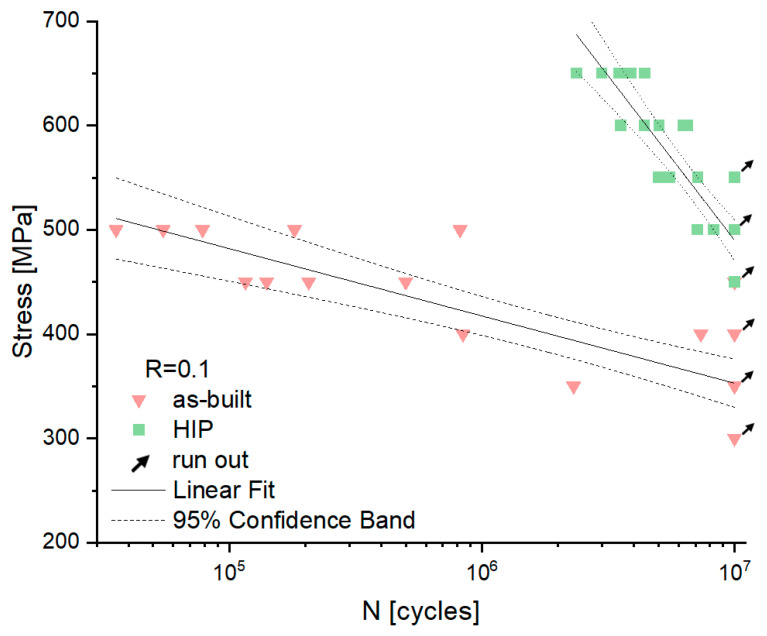
S–N curve fatigue results—relationship between the applied stress and the number of cycles to failure of the as-built, HIP specimens. Solid lines present regression fit, dotted lines present 95% confidence bounds from regression fit.

**Figure 15 materials-16-01201-f015:**
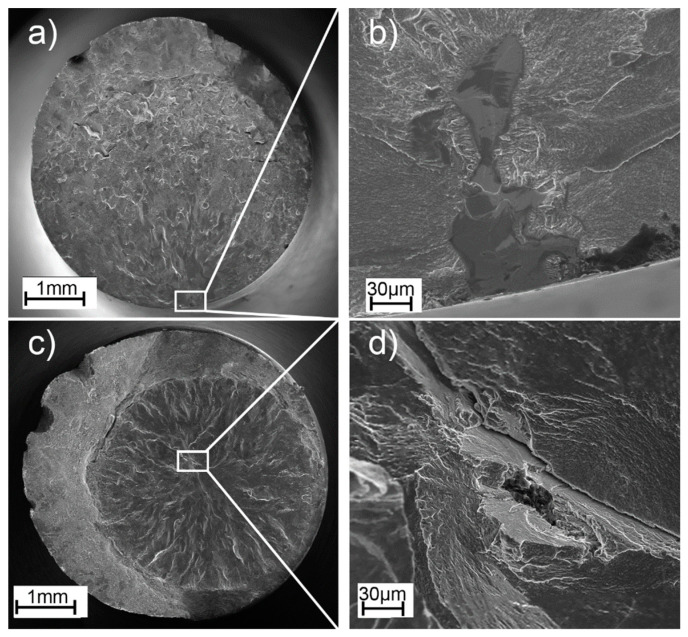
Representative SEM images of the fracture surfaces of components subjected to cyclic loading with the marked fracture initiation site: (**a**,**b**) as-built, applied stress of 500 MPa/819892 cycles; (**c**,**d**) HIP, applied stress of 650 MPa/2985343 cycles.

**Table 1 materials-16-01201-t001:** The chemical composition of Ti-55511 alloy according to the certificate provided by the manufacturer.

	Ti	Al	Mo	V	Cr	Fe
wt %						
**Reference**	Balance	4.4–5.7	4.0–5.5	4.0–5.5	0.5–1.5	0.5–1.5
**Powder**	Balance	4.92	5.09	4,97	1.02	1.02

**Table 2 materials-16-01201-t002:** Scan parameters for the XCT analysis.

Voltage (kV)	Current (µA)	Voxel Size (µm)	Prefiltration Cu (mm)	Int. Time (s)	Number of Projections
180	200	9,8	2	1.5	1050

**Table 3 materials-16-01201-t003:** The chemical composition of the Ti-55511, XRF analysis.

	Ti	Al.	Mo	V	Cr	Fe
	wt %					
**Reference**	balance	4.4–5.7	4.0–5.5	4.0–5.5	0.5–1.5	0.5–1.5
**Powder**	balance	4.92	5.09	4.97	1.02	1.02
**as-built**	balance	4.17	4.39	3.95	0.89	0.97
**HIP**	balance	4.21	4.15	4.01	0.93	0.92

**Table 4 materials-16-01201-t004:** Relative density of the specimens for the mechanical tests in the as-built and HIP state.

Specimens		As-Built	HIP
**ASTM E8M**	Material volume [mm^3^]	90.83	90.64
	Defect volume [mm^3^]	0.04	0
	Defect volume ratio [mm^3^]	0.05	0
	Relative density [%]	99.95	100
**ASTM E466**	Material volume [mm^3^]	87.21	87.46
	Defect volume [mm^3^]	0.05	0
	Defect volume ratio [mm^3^]	0.06	0
	Relative density [%]	99.94	100

**Table 5 materials-16-01201-t005:** Strength properties of the PBF-EB/Ti-55511 specimens in the as-built and HIP state.

	UTS [MPa]		YS [MPa]		ε [%]	
	As-Built	HIP	As-Built	HIP	As-Built	HIP
**Mean**	1060	856	1009	814	12.22	17.02
**SD**	19.84	5.30	23.94	8.17	2.84	1.45
**CV**	0.0187	0.0062	0.0237	0.0100	0.2326	0.0851
**Range**	45.94	14.08	52.58	17.97	6.71	2.99
**F-test, p**	0.57	0.93	0.24	0.33	0.70	0.06

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
