# Peer review of "The Effect of Postprocessing on the Fatigue Properties of Ti-5Al-5Mo-5V-1Cr-1Fe Produced Using Electron Beam Melting"

_materials, 2023, doi:10.3390/ma16031201_

Round 1

Reviewer 1 Report

This paper presents the results of research aimed at improving the fatigue properties of Ti55511 alloy by reducing the influence of lack of fusion defect on the fatigue damage. The effect of HIP on the reduction of pores and the degree of sphericity using XCT system was presented. The change in the microstructure after HIP was analyzed. Based on the criterion of journal, some parts of this work should be modified.

1) In the section of “Materials and Methods”, it is suggested to add a simple diagram of the experimental equipment or the experimental process for convenient reading.

2) At the end of 3.1, the hardness values of samples under the two conditions are only shown one average value. Which areas are tested? How many values have been obtained? Please show in detail or give an schematic diagram to indicate the test area. It is suggested to add the figure.

3) In the section of 3.3, some explanations can be added to the mechanical properties of the samples of the two series from the aspect of microstructure.

4) According to Table 3, after EBM process, not only Al, but also Mo and V evaporate in large quantities. Why only Al is mentioned. Please indicate.

5) Please analyze the reasons for the change in mechanical properties of the sample after HIP treatment from the perspective of microstructure.

Author Response

Thank you for your comments. I am sending the respond the attachment. Please see the attachment.

Kind regards,
Michal Karoluk

Reviewer 2 Report

Manuscript Number: Materials 2170265

Title:  Postprocessing effect on fatigue properties of Ti-5Al-5Mo-5V-2 1Cr-1Fe produced by Electron Beam Melting

By:  MichaÅ‚ Karoluk, Karol Kobiela, Marcin Madeja, Robert Dziedzic, Grzegorz ZióÅ‚kowski,Tomasz Kur-4 zynowski

Reviewer’s Comments to Authors:

  In this study, additive manufacture of Ti-5Al-5Mo-5V-1Cr-1Fe alloy (Ti-55511) is performed by using electron beam melting (PBF-EBM) process. In the as-built state, the sample has a relatively high density (99.95) but consists of a significant number of fine spherical pores. Besides, the defects related with the lack of fusion are unavoidable and difficult to characterize in the printed sample, which can obviously degrade the fatigue properties. The application of hot isostatic pressing (HIP) can reduces those fine pores and the discontinuities (lack of fusion). The experimental work was delicate and well described. The figure and table designs are clear. Overall, the results are noticeable and the conclusions are solid. As a reader, I would like to see more detail on the material used for this study. According to the present version, I put forward some comments as follows:

Q1: Obvious typing error Fr and XCT did not define in the text.     

Q2: The authors need to provide extra photos at low magnification to show the melt pool morphology of the inspected samples in the as-deposited and HIP conditions. Also, the reader may want to know where those fine pores locate along the melt pool boundary or within the nugget.      

Q3: Ti55511 alloy is a near β alloy. As mentioned in the text, the α phase content is about 40% in the as-built Ti55511 sample and about 80% in the HIP sample. The cooling rate is known to play an important role in the formation of α phase in the Ti55511 alloy. XRD pattern of the feedstock should be included in Fig. 4. Moreover, only vague feature to show the microstructure of the feedstock is present in Fig.2b. As a reader, I hope to know the exact microstructure of the feedstock and then compared to the as-built sample.

Author Response

Thank you for your comments. I am sending the respond the attachment. Please see the attachment.

Kind regards

Reviewer 3 Report

The authors have done work on an important and relevant topic. For a better perception of the article by readers, it would be good to make a few improvements.

In the introduction, there is practically no information about the state of the research on the topic under study. There are already quite a few works on EBM printing with titanium alloys. And in these works there is already information about the resulting porosity. There are also works on HIP processing. You should expand your review. Add links to these studies. Show why your work is relevant.

In the Methodology section, it would be good to indicate the equipment on which the mechanical tests were carried out. In parentheses indicate the manufacturer and country of manufacture of the equipment. Also indicate in what scheme the electron beam moved when printing samples. Printed in straight lines from edge to edge. Or each sample was printed in a circle separately.

Your article talks about a very significant change in the phase composition of the alloy during HIP processing. But in the article itself, very little attention is paid to the discussion of this fact. It would be necessary to explain the mechanism of such a transformation during HIP treatment. It is possible to give a state diagram and show how the pressure led to the appearance of another 40% of the α-phase. Also in conclusion 4 you explain that you have removed the lack-of-fusion type defect. But the lack-of-fusion type defect itself was not found. At the same time, bypass the fact of a change in the phase composition. You have reduced the amount of solid β-phase and added the amount of plastic α-phase. This will greatly affect the fatigue properties. These issues are best discussed in more detail in the article. It is possible to give a description of the properties of the α-phase and β-phase.

Author Response

Thank you for your comments. I am sending the respond the attachment. Please see the attachment.

Kind regards,

Round 2

Reviewer 1 Report

The format of the tables should be improved before publication.

Reviewer 3 Report

The authors eliminated the comments and finalized the article. The article can be published in my opinion.